# Involvement of Nitric Oxide in Protecting against Radical Species and Autoregulation of M1-Polarized Macrophages through Metabolic Remodeling

**DOI:** 10.3390/molecules28020814

**Published:** 2023-01-13

**Authors:** Junichi Fujii, Tsukasa Osaki

**Affiliations:** Department of Biochemistry and Molecular Biology, Graduate School of Medical Science, Yamagata University, Yamagata 990-9585, Japan

**Keywords:** NOS2, tricarboxylic acid cycle, urea cycle, aconitase, polyamines

## Abstract

When the expression of NOS2 in M1-polarized macrophages is induced, huge amounts of nitric oxide (•NO) are produced from arginine and molecular oxygen as the substrates. While anti-microbial action is the primary function of M1 macrophages, excessive activation may result in inflammation being aggravated. The reaction of •NO with superoxide produces peroxynitrite, which is highly toxic to cells. Alternatively, however, this reaction eliminates radial electrons and may occasionally alleviate subsequent radical-mediated damage. Reactions of •NO with lipid radicals terminates the radical chain reaction in lipid peroxidation, which leads to the suppression of ferroptosis. •NO is involved in the metabolic remodeling of M1 macrophages. Enzymes in the tricarboxylic acid (TCA) cycle, notably aconitase 2, as well as respiratory chain enzymes, are preferential targets of •NO derivatives. Ornithine, an alternate compound produced from arginine instead of citrulline and •NO, is recruited to synthesize polyamines. Itaconate, which is produced from the remodeled TCA cycle, and polyamines function as defense systems against overresponses of M1 macrophages in a feedback manner. Herein, we overview the protective aspects of •NO against radical species and the autoregulatory systems that are enabled by metabolic remodeling in M9-polarized macrophages.

## 1. Introduction

Endothelial cells play an essential role in acetylcholine-induced relaxation of the vasculature. Endothelial cell-derived relaxing factor (EDRF) refers to compounds that cause the relaxation of vascular smooth muscle cells. After the identification of nitric oxide (•NO) as an EDRF, nitric oxide synthase (NOS) was isolated and extensively characterized [1]. Several other enzymatic and nonenzymatic processes are now known to produce •NO from nitrite [2].

Cell-mediated immunity is achieved by cytotoxic T cells, natural killer (NK) cells, and macrophages. Upon stimulation by lipopolysaccharide (LPS) and inflammatory cytokines such as interferon-γ (IFN-γ), macrophages are polarized to the M1 subtype and protect against pathogens and parasites [3]. NOS2 is abundantly induced in M1-polarized macrophages and produces more •NO than from any other source and exerts antimicrobial effects (Figure 1). •NO derived from NOS2 also plays significant roles in antitumor activity and in nonimmune cells [4,5]. 

NADPH oxidase (NOX), notably NOX2, is also activated in M1 macrophages and produces superoxide (O_2_•^−^) that is then converted to other reactive oxygen species (ROS) [6]. •NO reacts with ROS to form reactive nitrogen oxide species (RNOS). Peroxynitrite (ONOO^–^), which is formed from •NO and O_2_•^−^, is a powerful oxidant and may be responsible for oxidative damage to surrounding tissues [7]. While •NO exerts potentially beneficial effects, accidentally produced RNOS may exhibit deleterious effects, which leads to a controversial assessment of the roles of NOS2 in vivo. Therefore, it would be wise to consider the primary actions of •NO separately from those of secondarily produced RNOS [8]. 

In this review article, we focus on the beneficial aspects of •NO produced by NOS against oxidative insult caused by oxygen radicals under conditions of inflammation. We also discuss metabolic remodeling that enhances M1 macrophage function and simultaneously activates its autoregulatory system to normalize overresponses.

## 2. Production and Reactions of •NO

NOS was the first enzyme to be identified that produces •NO using arginine as the nitrogen source [9]. It is now known that several enzymatic and nonenzymatic reductions produce •NO from nitrite/nitrate [2]. Essentially, all cells produce •NO, but the amounts and functions of the resulting •NO vary greatly depending on the type of cells or their redox environment. We briefly review the current state of our knowledge concerning •NO in this section. 

### 2.1. •NO Production by NOS and Other Pathways

Three NOS isoforms, namely the neuronal form (NOS1), the inducible form (NOS2), and the endothelial form (NOS3), produce •NO in response to corresponding physiological stimuli and exert a variety of functions to maintain health [10]. •NO is produced from arginine and molecular oxygen by NOS via a 6-electron reduction. NOS utilizes electrons from NADPH, similar to many other redox enzymes, but also requires additional cofactors: tetrahydrobiopterin, FAD, heme, and calmodulin. NOS1 and NOS3 are constitutively present and are activated by calcium-bound calmodulin upon the corresponding stimuli [9]. Readers are referred to early studies on mice with deficiencies of these isoforms, which mainly focus on cardiovascular function and renal pathophysiology, e.g., [11,12]. On the other hand, NOS2 expression is primarily induced in macrophages in response to inflammatory stimuli, and no further stimulation is required to activate NOS2 to produce •NO. 

In addition to NOS, other enzymes including xanthine oxidase, sulfate oxidase, and aldehyde oxidase convert nitrite to •NO [2]. In red blood cells, hemoglobin produces •NO through the reductive recycling of nitrite [13,14]. Nitrite/nitrate is also nonenzymatically reduced to •NO under acidic conditions in the stomach [15]. The fact that the mice lacking in all three NOS isozymes are viable [16] suggests that the compensatory actions of •NO originate from other sources. On the other hand, mice that have been bred under a deficiency of dietary nitrite and nitrate for a long time show unhealthy characteristics, such as the metabolic syndrome, endothelial dysfunction, and cardiovascular death [17]. These observations suggest that, along with •NO produced by NOS-catalyzed reactions, nitrite and nitrate are also essential sources for producing •NO and maintaining animal health.

### 2.2. Iron as a Target of •NO

Heme and nonheme iron are both preferential targets of •NO [10,18,19]. EDRF function is the primary action of •NO in vascular systems. NO produced by NOS3 in endothelial cells is transmitted to smooth muscle cells. The binding of •NO to heme in the soluble guanylate cyclase stimulates the catalytic conversion of GTP to cyclic GMP (cGMP). The resulting cGMP activates cGMP-dependent protein kinase, leading to vascular relaxation.

While carbon monoxide and oxygen molecule bind ferrous iron that is mostly present in the form of heme, •NO exceptionally binds ferric iron too [18]. The labile iron pool (LIP), which is defined as the iron fraction chelatable by high-affinity metal chelators, accounts for 0.1–3% of total cellular iron. The reaction of •NO with LIP produces stable dinitrosyl iron complexes, which are the most abundant adduct and may act as •NO carriers within cells [20,21]. 

•NO also preferentially reacts with the iron–sulfur (Fe-S) cluster that is commonly involved in electron transfer reactions. The Fe-S cluster is not as rigid as heme and is more sensitive to being destroyed by ROS/RNOS [22,23]. Among enzymes that are associated with the Fe-S cluster, aconitase in which 4Fe-4S plays an essential role is one of the most sensitive enzymes to ROS/RNOS [24,25,26]. While cytosolic aconitase 1 acts as a iron-regulatory protein (IRP), mitochondrial aconitase 2 (ACO2), a tricarboxylic acid (TCA) cycle enzyme, converts citrate to isocitrate. The oxidative inactivation of aconitase by anionic oxidants such as O_2_•^−^ and ONOO^–^ releases one ferrous iron unit along with hydrogen peroxide, which results in the formation of 3Fe-4S [23]. In fact, however, the •NO-dependent inactivation of ACO2 occurs slowly (~0.65 M^−1^ s^−1^), while ONOO^−^ inactivates it much faster (~1.4 × 10^5^ M^−1^ s^−1^) [27,28]. The release of iron from IRP stimulates its binding to iron responsible elements in some genes that are responsible for iron metabolism [29]. This ROS/RNOS-dependent structural change of IRP is reversible. When the 4Fe-4S cluster is reformed in the presence of free iron and reductants such as glutathione (GSH) (t_1/2_ = 12 min), the protein structure is converted into the active form with aconitase activity [30]. Regarding the mitochondrial isoform ACO2, a recent study reported that the •NO-mediated inhibition of ACO2 is responsible for the M1 polarization of the macrophage [31] as described below. Because heme and the Fe-S cluster are pivotal prosthetic groups for some proteins that are largely involved in electron transfer and redox reactions, the excessive production of •NO may exert cytotoxicity through causing iron-dependent enzymes to become dysfunctional [32].

### 2.3. Peroxynitrite as a Potent Oxidant

Activated macrophages produce a variety of inflammatory mediators, which include •NO, O_2_•^−^, and cytokines. O_2_•^−^ and •NO along with other ROS/RNOS act primarily as host defense systems against microbial infection but may also cause detrimental effects on surrounding tissues [33,34,35]. NOX is constitutively present and activated upon an infection and O_2_•^−^ is then produced to kill microbes. NOS2 expression is induced in macrophages, which results in the production of huge amounts of •NO [9]. The co-activation of NOX and NOS2 coordinately produce a variety of ROS and RNOS (Figure 2). •NO is a gaseous radical molecule and rapidly reacts with other radical species and biological compounds. Notably, the reaction between O_2_•^−^ and •NO proceeds in a diffusion-limited manner (k = 1.9 × 10^10^ M^−1^s^−1^) [36], and the resulting ONOO^–^ is a strong oxidant [7,37]. O_2_•^−^ possess a negative charge, which prevents it from permeating through lipid bilayers. While O_2_•^−^ diffuses only for a short distance within cells, •NO readily diffuses through lipid bilayers and can function in intercellular signaling. Accordingly, ONOO^−^ tends to be formed at the site where O_2_•^−^ is produced, dominantly in mitochondria [38]. 

In vivo ONOO^−^ formation is estimated to be 0.1 to 0.5 µM s^−1^ and its steady-state concentration is ~1 nM [39]. However, the production of ONOO^−^ in macrophages is as high as 50–100 µM min^–1^ [40]. ONOO^−^ is unstable with a half-life of ~10 ms at physiological pH because protonated form ONOOH (pKa = 6.8) is immediately isomerized to nitrate in aqueous conditions. Based on these properties, the distance of diffusion is assumed to be ~10–20 µm inside cells. During the decomposition of ONOOH, hydroxyl radicals (•OH) and nitrogen dioxide radicals (•NO_2_) may be produced under a hydrophobic environment such as inside a membrane bilayer [41]. ONOO^−^ interacts rapidly with carbon dioxide (CO_2_) (k = 4.6 × 10^4^ M^−1^ s^−1^) and is consequently converted to carbonate radical (CO_3_•^−^) + •NO_2_, which also exerts bactericidal effects in phagosomes, or CO_2_ + NO_3_^−^ [42,43]. Several enzymes such as GPX1 [44], PRDX2 (k = 1.4 × 10^7^ M^−1^ s^−1^ at 25 °C, pH 7.4) [45,46], and mitochondrial PRDX3 (1 × 10^7^ M^−1^ s^−1^ at 25 °C, pH 7.8) [47] may detoxify ONOO^−^. Nevertheless, ONOO^−^ is considered to be a major oxidizing RNOS in vivo situation [48]. 

### 2.4. •NO-Mediated Modification of Molecules

Organic compounds experience two types of reaction with •NO, i.e., nitration and nitrosylation. Nitration results in the formation of nitro compounds (RNO_2_), which are relatively stable and hence can be useful as biomarkers [49,50]. Nitrosylation occurs in carbon, sulfur, and oxygen in biological compounds (RCH, RSH, and ROH), which results in the production of RCNO, RSNO, and RONO, respectively. These compounds with •NO-mediated modification may show differential responses in biological systems. 

Among the RNO_2_ species, the formation of 3-nitrotyrosine (3-NT) is an amino acid modification that occurs frequently and has attracted the attention of researchers [51]. 3-NT is produced under conditions of nitrosative stress with excessive RNOS and plays roles in cardiovascular dysfunction [52], neurological disorders [53], and diabetes [54]. ONOO^−^ is a major reactant in the formation of 3-NT. For example, tyrosine nitration has been observed in ACO2 in several animal models with inflammation such as sepsis and diabetes [38]. Two specific 3-NT species that adjust the active site have been reported, but nitration does not affect the enzymatic activity of ACO2. Although the nitration of tyrosine may cause alterations in the structure and function of certain proteins, it is more useful as a molecular footprint for assessing •NO production and nitrosative stress [52]. Nitration also occurs in bases of nucleic acids, as represented by 8-nitroguanine in DNA [55,56] and 8-nitroguanosine 3’,5’-cyclic monophosphate (8-nitro-cGMP) [57], may exert unique roles in pathogenesis such as in cancer [58]. 8-nitro-cGMP uniquely induces autophagy, which excludes invading group A *Streptococcus* from a cell [59].

While the RSNO formation by •NO is rather efficient process, enzymes designated as SNO synthases may accelerate the formation of SNO as reported in *E. coli* [60]. RSNO undergoes several conversion reactions depending on the reaction conditions [61]. Homolytic bond break converts RSNO to RS• + •NO. The reaction of R_1_SNO with the other thiol compound (R_2_SH) produces either R_1_SH + R_2_SNO by transnitrosation or R_1_SSR_2_ + HNO. Because of its ability to transfer NO to other thiols, SNO can act as a signaling molecule, which is well-recognized in cardiovascular systems [62]. S-nitrosylated hemoglobin may be a critical mediator of blood flow in hypoxic tissues [63]. Because GSH is the most abundant thiol within cells, nitroso glutathione (GSNO) is a dominant RSNO in •NO-producing cells [64]. GSNO is a donor for transnitrosylation reactions and acts as a central intermediate in the formation and degradation of cellular RSNO [65,66]. Proteins that possess reactive cysteine are dominantly nitrosylated in activated macrophages [67,68]. Through altering the functions of target proteins, nitrosylation may cause dysfunctions, aberrant activation of physiological processes, and ultimate cell death [50].

Either suppression of •NO production or denitrosylation may be potential therapeutics for the treatment of certain diseases caused by nitrosative stress that are associated with excessive •NO production [32,69]. GSNO is reductively denitrosylated by several proteins in an NAD(P)H-dependent manner and produces ammonia and oxidized GSH (GSSG) [70]. These enzymes include carbonyl reductase 1 (CBR1) and class III alcohol dehydrogenase (ADH5) [50,71]. On the other hand, thioredoxin/thioredoxin reductase systems degrade GSNO to •NO + GSH/GSSG. In fact, GSNO is the preferred substrate for CBR1 among the known substrates with carbonyl groups. Coenzyme A (CoA), an important nonprotein thiol, is subject to nitrosylation. *S*-Nitroso-CoA may mediate protein *S*-nitrosylation [72]. As a result, a defect in the denitrosylation of *S*-Nitroso-CoA by the genetic ablation of the responsible gene AKR1A impairs glycolysis in the reaction catalyzed by pyruvate kinase 2, which results in an elevation in NADPH production via the stimulation of glucose-6-phosphate flow to the pentose phosphate pathway [73].

### 2.5. •NO as a Potent Radical Scavenger

Because the sustained presence of O_2_•^−^ increases the risk of production of more harmful radicals such as hydroxyl radicals [22], the removal of O_2_•^−^ by •NO at an early stage may prevent oxidative injury by outweighing the toxicity of ONOO^−^ [74,75]. Alveolar macrophages from NOS2-knockout mice show higher levels of O_2_•^−^ and hydrogen peroxide compared to the levels in wildtype mice [76], which implies the actual scavenging O_2_•^−^ by •NO in vivo. We recently showed that both intrinsically produced •NO and long-lasting •NO-donor compounds exert protection against oxidative damages caused by O_2_•^−^ in LPS-treated SOD1-knockout macrophages [77]. Primary macrophages isolated from mice with a double deficiency of NOS2 and SOD1 are the most vulnerable compared to singly deficient mice. These observations suggest that, despite the strong oxidant potential of the resulting ONOO^−^, low stability at physiological pH and the presence of reducing enzymes may collectively overcome the cytotoxic effects of ONOO^−^ in macrophages. Thus, from a different point of view, ONOO^−^ is not always deleterious but protective in activated macrophages and some other cells expressing NOS2. 

•NO also exhibits a high reactivity for lipid radicals (L• and LOO•). The reaction of •NO with such radicals is in fact quite rapid, with rate constants of 1–3 × 10^9^ M^−1^s^−1^ [78], which results in the formation of LNO and LOONO [8] (Figure 3). The resulting nitroso lipids are much less toxic compared to lipid radicals, so that these reactions alleviate oxidative damage caused by lipid radicals. For example, in liposome systems, the formation of phosphatidylcholine and cholesterol peroxides is actually suppressed by •NO [79,80]. Several •NO donors have been reported to inhibit the macrophage-dependent oxidation of low density lipoprotein (LDL) in cultured cells [81]. While NOS2-deficient macrophages show enhanced oxidation in LDL [82], the pretreatment of oxidized LDL with •NO decreases apoptosis-inducing ability with suppressed lipid peroxyl radical formation [83].

A lipid peroxidation product (LOOH) is a direct executer of ferroptosis, a type of iron-dependent necrotic cell death [84,85]. Ferroptosis is particularly problematic in cases of inflammation in which activated macrophages play an important role in pathogenesis [86]. While it is known that vitamin E (α-tocopherol) suppresses lipid peroxidation and subsequent ferroptosis by eliminating radical electrons, the rate of reactions of •NO with lipid radicals is two to three orders of magnitude higher than that of vitamin E [87]. Thus, •NO is considered an efficient terminator of the radical chain reaction that is operative during lipid peroxidation reactions, even in the presence of vitamin E [88]. •NO reportedly inhibits lipoxygenase (LOX)-mediated lipid peroxidation [24,89]. A subsequent study reported that the •NO-mediated suppression of lipid peroxidation is not due to the inhibition of LOX but, rather, the termination of the chain reaction by •NO [90]. Because LOX-induced lipid peroxidation plays primary roles in executing ferroptosis, these observations explain the antiferroptotic function of •NO. A long-lasting •NO donor effectively rescues hepatoma-derived cells from death that is induced by several ferroptotic stimuli: cultivation in cysteine-deprived medium and the inhibition of either the cystine transporter xCT, or the LOOH-reducing enzyme GPX4 [91]. These results suggest that, in addition to •NO produced by NOS2, exogenous •NO suppresses ferroptosis via termination of the radical chain reaction. The nitrosylation of phosphatidylethanolamine in activated macrophages leads to the suppression of ferroptosis under pro-inflammatory conditions [92,93]. Moreover, *Pseudomonas aeruginosa*-stimulated ferroptosis in epithelial cells can be prevented by macrophage-derived •NO [94]. Thus, •NO suppresses cell death in macrophages and other cells under differential ferroptotic stimuli by terminating radical chain reactions and, hence, suppressing the production of LOOH. 

## 3. Metabolic Remodeling and Autoregulation of M1-Polarized Macrophages

In response to inflammatory stimuli such as IFN-γ and LPS, macrophages are polarized to the M1 type with remodeled metabolic pathways and an altered gene expression [95]. In spite of the fact that there are multiple sources for •NO in vivo, NOS2-derived •NO in M1-poralized macrophages is considered to play a dominant role in inflammatory diseases. Notably, in addition to a cytokine storm, excessively produced ROS and RNOS cause the deterioration of sepsis, which is defined as a microbial infection with organ dysfunction [96,97].

### 3.1. Antithetical Action of •NO under Inflammation

The NOS2 expressed in M1 macrophages produces large amounts of •NO that appear to be primary protectors against microbial infection [98,99]. It is thought that ROS and RNOS coordinately exhibit innate immune defense to microbes such as *Leishmania donovani* and *Mycobacterium tuberculosis* [34,100]. However, •NO also exhibits antithetical functions, both beneficial and deleterious reactions to host animals. Under conditions of sepsis, excessive immune responses occur upon microbial infections that are sufficiently serious to impair host organs, such as the cardiovascular system, the lung, and the brain [101]. The production of high levels of ROS and RNOS as well as inflammatory cytokines aggravates the progression of microbial infections. Investigations of genetically modified mice indicate the existence of a pathophysiological role for NOS2 in vivo, e.g., [76,102,103]. It has also been reported that these mice exhibit phenotypically dissimilarities in some issues, suggesting that the exact evaluation of the action of •NO in an in vivo situation is a difficult task. Roughly speaking, it is conceivable that •NO primarily exerts a beneficial action such as cellular signaling and bactericidal action, while secondarily formed RNOS may affect various influences depending on their mode of production.

Calcium ions are the principle activators for NOS1 and NOS3 via binding to calmodulin, but NOS2, which contains bound calmodulin, is constitutively active. Accordingly, the production of •NO by NOS2 is mainly regulated through gene expression and the degradation of the NOS2 protein by proteasomes [104,105]. NF-*κ*B and IRF1 are the main transcriptional regulatory factors for the induction of NOS2 and, as a result, their inhibition suppresses NOS2 induction [106]. NOS2 is also induced in stimulated vascular smooth muscle cells and produces substantial amounts of •NO [107]. •NO itself has been reported to negatively regulate the expression of NOS2 by suppressing the cytokine-induced activation of NF-*κ*B via blocking its phosphorylation in rat vascular smooth muscle cells [108]. The NOS2 protein is not expressed in most cells, including macrophages, in the absence of inflammatory stimuli, but surprisingly, the kidney medulla expresses NOS2 constitutively [109,110]. The regulation of blood pressure, cholesterol levels, and sensitivity to salt may be attributed to the formation of •NO derived from constitutive NOS2 in the kidney [111], but data concerning this issue vary depending on the report [12,112]. There are also a few factors that could also regulate NOS2 activity post-translationally, such as interactions with heat shock protein 90 (hsp90) as an allosteric enhancer [113] and *N*-glycosylation as a suppressor [114]. 

In addition to antimicrobial effects, excessive ROS and RNOS produced in M1 macrophages may impair the surrounding host tissue. NOS2 consists of two functional domains: an *N*-terminal oxygenase domain and a *C*-terminal reductase domain. Tightly bound calmodulin links these domains, which enables electron transfer from NADPH to the substrate arginine [9]. The presence of excessive levels of RNOS may cause uncoupling between these two domains and produce O_2_•^−^ from NOS2, which aggravates organ damage, as is presumed for cardiovascular dysfunction under sepsis [115]. Macrophages themselves are relatively resistant to these oxidizing molecules ROS/RNOS and ferroptotic death under a xCT deficiency [116,117]. The proliferation of microbes requires iron as an essential nutrient. To defend against microbial proliferation inside cells, macrophages have iron exclusion systems: the lysosomal iron pump protein natural resistance-associated macrophage protein 1 (Nramp1) and the iron export protein ferroportin [117]. It has been suggested that these iron-excluding systems may render macrophages resistant to ferroptosis in which free ferrous iron plays essential roles. 

### 3.2. Metabolic Remodeling of the TCA Cycle

M1-polarized macrophages characteristically exhibit phenotypes in which the synthesis of cytokines and lipids and the production of O_2_•^−^ and •NO are stimulated (Figure 4A), e.g., [118,119,120]. ROS/RNOS affects metabolism via the TCA cycle and electron transport complexes by interfering with the activities of Fe-containing enzymes in mitochondria [121,122]. Notably, as described above, ACO2 is highly susceptible to oxidative modification via several ROS/RNOS-involved processes [22,23,123]. The protease proteolytically eliminates oxidized and damaged proteins in mitochondria, and ACO2 has been identified as one of major target molecules of this enzyme [124].

As a result of the inhibition of ACO2 activity, other pathways that are involved in carbon metabolism are also altered substantially [125,126,127,128,129]. Inhibiting the conversion of citrate to isocitrate causes the accumulation of citrate, which is then recruited for the synthesis of fatty acids. Iso-aconitate, which is an intermediate compound that is produced during the conversion from citrate to isocitrate by ACO2, appears to accumulate. Iso-aconitate decarboxylase (ACOD1), also referred to as the immune reactive gene 1 (IRG1), is induced specifically in M1 macrophages and decarboxylates iso-aconitate to itaconate [130,131]. The resulting itaconate participates in multiple reactions that primarily involve protecting host animals against oxidative stress and microbial infections. Among them, itaconate inactivates succinate dehydrogenase (SDH), leading to the conversion of succinate to fumarate being attenuated. The accumulated succinate competes with 2-ketoglutarate (2-KG) for the prolyl hydroxylase reaction, which results in the degradation of the hypoxia-inducible factor (HIF)-1α by proteasomes to be suppressed and this event is not affected by oxygen conditions [132,133]. Stabilized HIF-1α is then translocated to the nucleus where it induces responsive genes that include IL-1β and enzymes in glycolysis [134,135]. 

The levels of glucose-6-phosphate increase as the result of elevated glycolysis, and a part of it flows to the pentose phosphate pathway, which is also stimulated by LPS treatment [134]. Glucose 6-phosphate dehydrogenase produces NADPH, which donates electrons to a variety of reactions including the production of •NO and O_2_•^−^ by NOS2 and NOX, respectively, and lipogenesis. NADPH is utilized for the biosynthesis and recycling of tetrahydrobiopterine, which is also a cofactor for the NOS reaction [136]. Thus, the remodeling of the TCA cycle is associated with the production of •NO and O_2_•^−^ as well as inflammatory cytokines.

On the other hand, a number of antioxidative systems commonly utilize NADPH for the reductive recycling redox molecules such as GSH and thioredoxin [137,138]. Nrf2 is a master regulator of protective genes against various types of stress, including oxidative stress [139]. While Nrf2 in the cytosol is largely associated with KEAP1, which stimulates the proteasome-mediated degradation of Nrf2, the elevation of ROS/RNOS results in the dissociation of KEAP1 and this, in turn, results in the stabilization of Nrf2. On the other hand, itaconate stabilizes Nrf2 by alkylating KEAP1, which eventually alleviates M1 macrophage-associated tissue injuries [129,140]. Thus, while the metabolic remodeling of the TCA cycle is responsible for phenotypic alterations, M1 macrophages are coordinately preparing self-protection systems against the produced ROS/RNOS.

### 3.3. •NO and Polyamines as the Amino Acid Metabolism-Associated Mediators

A causal connection between •NO production and the urea cycle has been an issue of interest, notably in macrophages, because they consume a large body of arginine for •NO production and also produce citrulline (Figure 4B) [141,142]. The uptake of arginine is aggressively stimulated in M1 macrophages under inflammatory conditions. The expression of the cationic amino acid transporter (CAT) 2 that constitutes system Y^+^ is induced and is responsible for the uptake of arginine [143]. On the other hand, the citrulline transporter remains ambiguous, although the involvement of SLC6A19 and SLC7A9 has been implied [144]. 

In the urea cycle of the liver, ornithine is converted to citrulline by the action of ornithine transcarbamylase (OTC), which, along with carbamoyl phosphate synthetase I, constitutes primary process for the detoxification of ammonia. However, the expression of OTC is limited to only a few organs and is nearly negligible in immune organs such as the thymus and the spleen [145]. Hence, the conversion of ornithine to citrulline is unlikely in macrophages and other OTC-deficient organs. Argininosuccinate synthetase is co-induced with NOS2 in M1 macrophages and accelerates the conjugation of citrulline with aspartate to form argininosuccinate [146]. Argininosuccinate is then broken down to arginine and fumarate by the action of argininosuccinate lyase. Thus, the amino group of aspartate along with nitrogen from citrulline become nitrogen donors for the urea, while the nitrogen of •NO is from the amino group of aspartate.

Arginase (ARG) is encoded by two genes, ARG1 and ARG2, and catalyzes the conversion of arginine to the urea and ornithine. Although NOS2 and ARG are assumed to be competing for the utilization of arginine, both ARG1 and ARG2 are actually co-induced with NOS2 in the macrophages upon inflammatory stimuli [147,148]. Given that the K_m_ values of arginases are ~10 mM, which is approximately three orders higher than that of NOS2 (K_m_ = ~5 µM) [148], the consumption of arginine by ARG may exert only a limited influence over NOS2 activity [149]. Nevertheless, either the inhibition of ARG2 activity or the genetic knockdown/ablation of the ARG2 gene enhances •NO production by increasing the levels of the NOS2 protein in macrophages [150,151]. These observations suggest the presence of regulatory machinery for producing •NO in which downstream metabolites of the ARG reaction appear to be involved. 

The induction of ARG1/ARG2, along with the lack of OTC, leads to the accumulation of ornithine in M1 macrophages upon inflammatory stimuli [141]. In addition to serving as a precursor for citrulline, ornithine is used for the synthesis of polyamines [152]. Ornithine decarboxylase (ODC) is the rate-determining enzyme for polyamine synthesis and converts ornithine to putrescine, which is further converted to spermidine and finally spermine. Polyamines play important roles in diverse biological processes, which include cell growth, differentiation, transformation, and apoptosis [153,154]. Members of P-type ATPase enzymes, the solute carrier (SLC), and ATP-binding cassette (ABC) proteins reportedly act as polyamine transporters [155]. Because polyamines support the proliferation of cancer cells, an understanding of the pathway for the synthesis of polyamines and their cellular transport may be an effective form of cancer therapy [156]. 

In the immune system, polyamines negatively regulate inflammatory responses [157,158], which may, in part, be associated with suppression of •NO production in macrophages [159,160]. Spermine suppresses •NO production during *H*. *pylori* infections, which is not attributed to the direct inhibition of NOS2 activity but, rather, the inhibition of translation from the mRNA [160]. On the other hand, polyamines inhibit the uptake of arginine by CAT2 without affecting the levels of the mRNA or the corresponding protein, which is most likely performed via competition with arginine [161]. A subsequent study showed that the epigenetic modification of histones by putrescine suppresses macrophage activation and M1 phenotypes that include •NO production [162]. Consistent with the immune-suppressive action of polyamines, ODC-deficient macrophages show a decrease in polyamine levels but an increase in the NOS2 activity [162]. Thus, it appears that polyamines downregulate NO production in M1 macrophages based on several mechanisms and may alleviate inflammatory insults. 

### 3.4. Metabolic Crosstalk between Remodeled TCA Cycle and Urea Cycle 

Regarding the remodeling of these metabolic pathways, we hypothesize that a novel metabolic linkage between the fragmented TCA cycle and incomplete urea cycle exists (Figure 5). Fumarate, which is originated from argininosuccinate along with phenylalanine and tyrosine, enters the TCA cycle and is converted to oxaloacetate via malate. Citrate synthase conjugates oxaloacetate with an acetyl group from acetyl-CoA to form citrate, which is recruited for lipogenesis or the production of itaconate by ACOD1. However, under conditions where pyruvate dehydrogenase is inhibited by •NO [31], oxaloacetate tends to be converted to aspartate by accepting an amino group from glutamate by means of aspartate aminotransferase (AST). Given that glutamate is a substrate that accepts an amino group from many other amino acids, these amino acids can become a nitrogen source for aspartate. In the urea cycle of the liver, the origin of one of the nitrogens in the resultant urea is ammonia, caused by the action of OTC coupled with carbamyl phosphate synthetase I, while the other nitrogen is from aspartate. However, in the case of an OTC deficiency, citrulline is not reconstituted from ornithine and ammonia-derived nitrogen. On the contrary, a recent study demonstrated that citrulline is cleaved to ornithine and isocyanate by means of the isocyanic acid synthetase activity of laccase domain-containing 1 protein (LACC1), which is induced by inflammatory stimuli in macrophages [163]. As a result, one nitrogen comes from aspartate, similar to the hepatic urea cycle, but the other comes from citrulline, which does not arise from ornithine but from arginine by means of NOS2. This causes the net consumption of citrulline and arginine, which explains the induction of CAT2 in M1 macrophages and the stimulated uptake of arginine from extracellular sources. 

The low K_m_ for arginine allows for the continuous production of •NO by NOS2, even under low concentrations of arginine. This implies that the stimulated uptake of arginine through the induction of CAT2 supports the robust production of ornithine and urea by ARG. The increase in ornithine, which is accomplished by the induced ARG and isocyanic acid synthetase, provides an advantage for the production of polyamines and, hence, cellular defense against excessive immune responses under conditions of inflammation [153,163]. Concerning the roles of urea, urease in *H. pylori* degrades urea to ammonia, thus allowing it to survive in gastric fluids by neutralizing the acidic environment, e.g., [164,165]. However, the physiological significance of urea produced in M1 macrophages remains obscure.

## 4. Perspectives

ONOO^−^ resulting from •NO and O_2_•^−^ is highly toxic; however, the beneficial action of scavenging O_2_•^−^ by •NO may outweigh such disadvantage in certain situations. Lipid peroxidation is enhanced by radical chain reactions and is associated with cellular dysfunction, such as ferroptosis, and a variety of diseases. The elimination of lipid radicals by interactions with •NO interferes with this chain reaction, and hence suppresses ferroptosis. Thus, the radical nature of •NO may contribute to defense against oxidative insults via scavenging toxic radical species, despite the fact that some RNOS are largely deleterious. Investigation of •NO function in terms of this issue is just beginning. When a methodology for in vivo evaluation of ferroptosis is established, the importance of antioxidation through radical scavenging by •NO will become unveiled in inflammatory diseases.

The TCA cycle and urea cycle are remodeled in M1 macrophages under conditions of inflammation. There is crosstalk between these metabolic pathways. Itaconate and polyamines that are produced from the remodeled TCA cycle and urea cycle, respectively, function to autoregulate M1 macrophage functions and host defenses against microbial infections. Further clarification of this type of metabolic remodeling would lead to a better understanding of the overall function of M1 macrophages and controlling them in cases of inflammatory diseases. Although induction of NOS2 occurs in some other cells, its relationship with metabolic systems, which has been revealed in M1 macrophages, remains largely unknown. Further elucidation of the role of NOS2-derived •NO in relation to other metabolic systems in a wide range of cells will lead to a comprehensive understanding of redox systems in vivo.

## Figures and Tables

**Figure 1 molecules-28-00814-f001:**
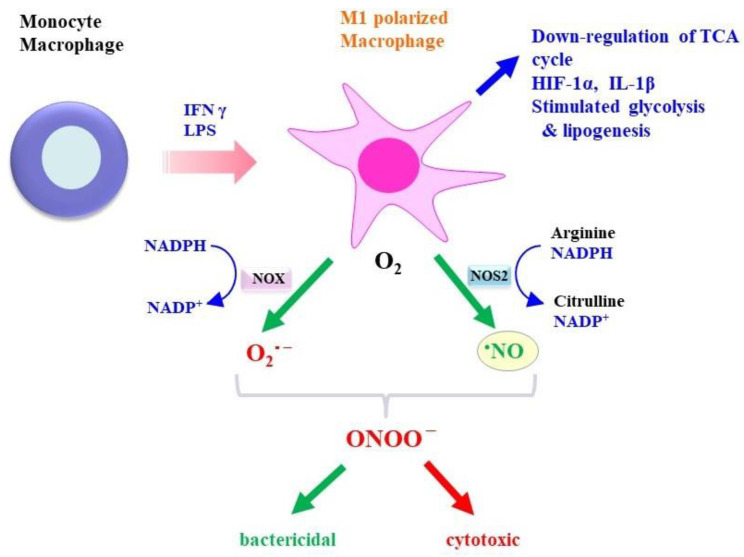
Properties of M1 macrophages: Upon stimulation by IFN γ and LPS, macrophages are polarized to the M1 type and produce O_2_•^−^ and •NO via NOX and NOS2, respectively, as well as inflammatory cytokines and other metabolites. Activation of HIF-1α and the stimulation of glycolysis and lipogenesis are also characteristics of M1 macrophages. ONOO^−^ is produced from O_2_•^−^ and •NO and may exert strong antimicrobial and tumoricidal action, but also cytotoxic action.

**Figure 2 molecules-28-00814-f002:**
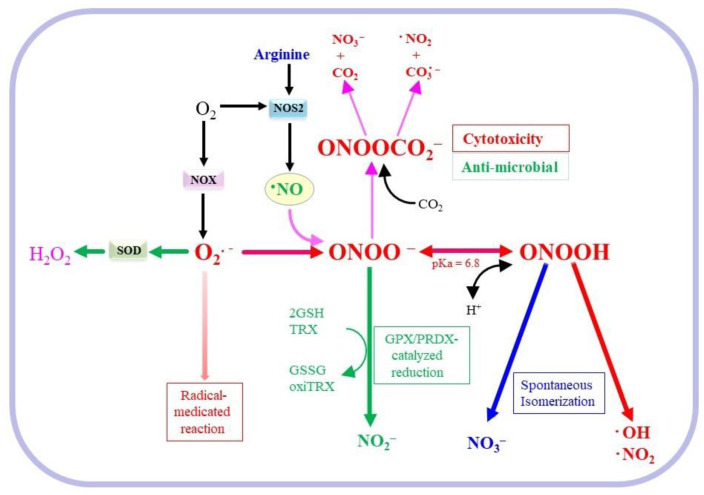
Conversion of •NO and ONOO^−^. Activated macrophages simultaneously produce O_2_•^−^ and •NO. ONOO^−^ is largely protonated to ONOOH (pKa = 6.8) at physiological pH and is spontaneously isomerized to nitrate. During the isomerization, deleterious radical species, •OH and •NO_2_, are produced in lipophilic environments. Cells have protective systems, glutathione peroxidase (GPX) and peroxiredoxin (PRDX), that reduce ONOO^−^ to nitrite, in GSH- and thioredoxin (TRX)-dependent manners, respectively. The reaction of ONOO^−^ with CO_2_ produces either CO_3_•^–^ + •NO_2_, which also have antimicrobial effects, or CO_2_ + NO_3_^–^.

**Figure 3 molecules-28-00814-f003:**
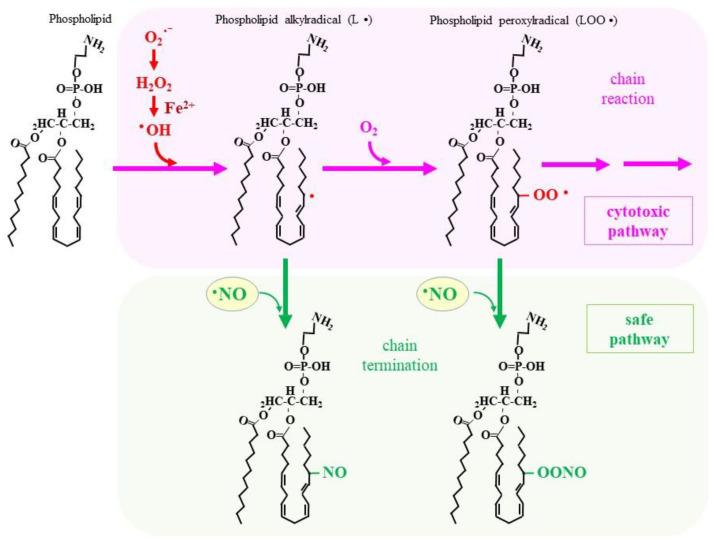
•NO suppresses ferroptosis by terminating the radical chain reaction of phospholipid radicals. •NO preferably reacts with L• and LOO• to form relatively stable C-nitroso compound (LNO) and O-nitroso compound (LOONO), respectively. As a result, the chain reaction is terminated, which eventually suppresses ferroptosis and other lipid peroxide-associated cell damages.

**Figure 4 molecules-28-00814-f004:**
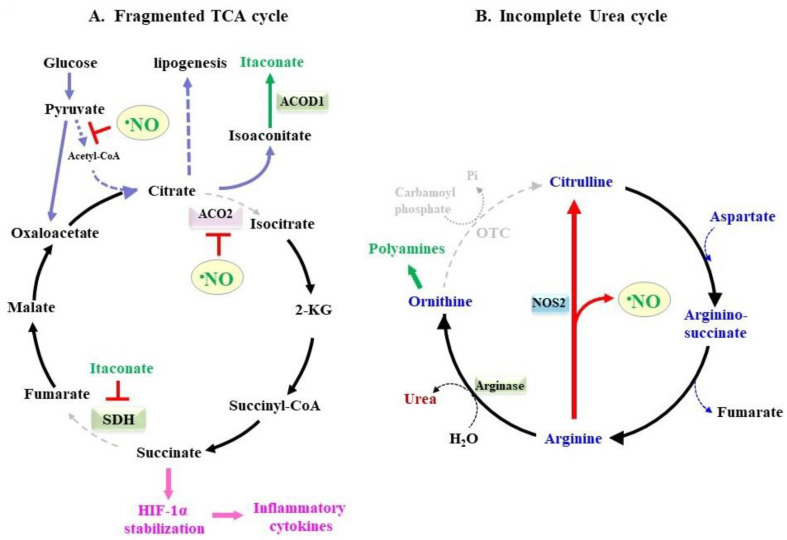
Metabolic remodeling of TCA cycle and urea cycle. (**A**) The TCA cycle in M1 macrophages is fragmented in the reaction catalyzed by ACO2, which leads to the flow of carbon to lipogenesis and the production of itaconate by means of ACOD1. ROS and RNOS are involved in the suppression of ACO2. The resulting itaconate inhibits SDH, which leads to the activation of HIF-1α and the subsequent induction of genes for inflammatory cytokines and glycolysis. (**B**) •NO is produced from arginine by NOS2 and results in citrulline. Since macrophages do not express OTC, ornithine is not utilized for citrulline synthesis, but is recruited for the synthesis of polyamines.

**Figure 5 molecules-28-00814-f005:**
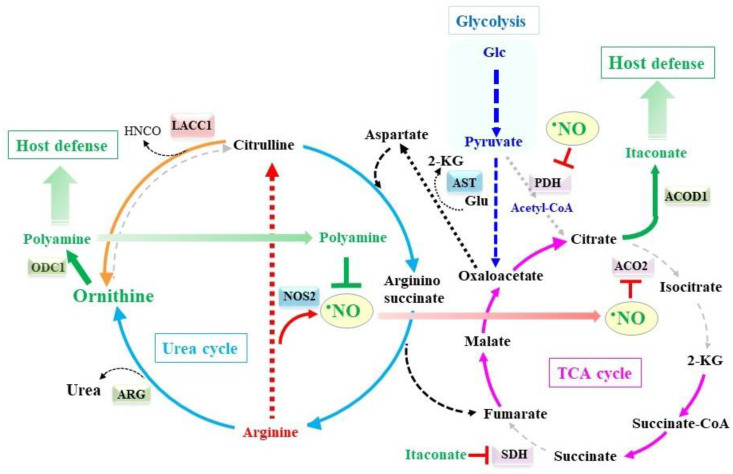
Crosstalk between the remodeled TCA cycle and urea cycle. The donation of nitrogen to citrulline from an amino group via argininosuccinate converts aspartate to fumarate. The resulting fumarate enters the TCA cycle and is converted into oxaloacetate through malate. Aspartate amino transferase (AST) then transfers an amino group from glutamate (Glu) to oxaloacetate and regenerate aspartate. Glutamate is regenerated by transferring an amino group from other amino acids. In total, the origin of urea nitrogen is amino acids, and none come from ammonia. LACC1 converts citrulline to ornithine and isocyanate by the action of isocyanic acid synthetase. Polyamines are synthesized from both citrulline and arginine via ornithine and suppress the production of •NO. As a result, overresponses of M1 macrophages, which are enabled by the remodeled TCA cycle, tend to be normalized by polyamines that are originated from the remodeled urea cycle.

## Data Availability

Not applicable.

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
