# Peer review of "Involvement of Nitric Oxide in Protecting against Radical Species and Autoregulation of M1-Polarized Macrophages through Metabolic Remodeling"

_molecules, 2023, doi:10.3390/molecules28020814_

Round 1

Reviewer 1 Report

The review paper is well written and discusses different production and reactions of nitric oxide as well as nitric oxide roles in M1-polrized macrophages. I have few comments before accepting the paper. 

Comments

- In introduction: It will be preferred if authors can add about the immunology cells' role in the  (macrophages, cytokines, T cell, NK cells ...etc.).

- Adding M1-polrized macrophage figures can be helpful to illustrate author findings.   

Minors

p.55 "NO" is the nitric oxide synthase?  

p.269 missing "the".

Author Response

The review paper is well written and discusses different production and reactions of nitric oxide as well as nitric oxide roles in M1-polrized macrophages. I have few comments before accepting the paper. 

Comments

- In introduction: It will be preferred if authors can add about the immunology cells' role in the  (macrophages, cytokines, T cell, NK cells ...etc.).

Responses:

Thank you for kind advice. This review article is written for the special issue of NO, so the immune system other than macrophages was not mentioned. Now we have added following brief statement on roles of representative immune cells in response to the comment.

“Cell-mediated immunity is achieved by cytotoxic T cells, natural killer (NK) cells, and macrophages. Upon stimulation by lipopolysaccharide (LPS) and inflammatory cytokines such as interferon-γ (IFN-γ), macrophages are polarized to M1 subtype and protect against pathogens and parasites [3]”

Adding M1-polrized macrophage figures can be helpful to illustrate author findings.  

Responses:

Fig. 1 schematically presents properties of M1-polarized macrophages. Because the special issue is dedicated to NO, but not immunology, we believe that this scheme will be sufficient for understanding properties of the M1-polarized macrophages.

Minors

p.55 "NO" is the nitric oxide synthase?  

Responses:

Thank you for pointing out the typo. We have corrected "NO" to "NOS".

p.269 missing "the".

Responses:

Thank you for pointing out the typo. We have corrected to "the".

Reviewer 2 Report

 The manuscript reviews the biological production and reactions of nitric oxide (NO). It is also discussed the influences of NO production in the tricarboxylic acid and urea cycles. This theme is interesting and may attract the interest of scientists working in the area of NO. Despite that, the manuscript could benefit from some corrections.

Comments:

Line 38 – “Peroxynitrite (ONOO–), which is formed from •NO and O2•−, is the most powerful oxidant and may be responsible for oxidative damage to surrounding tissues”. This sentence needs revision. The •OH is the most powerful (and less selective) known biological oxidant, not peroxynitrite.

In the section “Iron as a target of •NO” the authors correctly affirm that heme and non-heme iron are preferential targets of •NO. However, the non-heme includes not only the iron-sulfur clusters but also the labile iron pool (LIP). LIP is an important NO target and should be discussed in this section.

Lines 101-103 “The oxidative inactivation of aconitase releases one ferrous iron unit along with hydrogen peroxide, which results in the formation of [3Fe-4S] [21].” In this sentence the authors do not identify the oxidant that leads to aconitase inactivation, therefore it is not possible to affirm that hydrogen peroxide is a byproduct. Please, revise this information.

In the section “Peroxynitrite as a potent oxidant” the authors neglect the peroxynitrite’s pKa (6.8) and also the difference in reactivity of ONOOH and ONOO- species. The ONOOH decay leads to nitrate, •NO2 and •OH, not ONOO- decay. Please, correct this information (lines 144-148).

Line 177 – The authors should better explain why the RSNO formation by •NO is a slow process. Which mechanism are they referring to?

Line 179 – “Because RSNO may release bound •NO or NO+ and leave the disulfide bond in the original compound, it is not an end product but is regarded as a •NO donor.”

This referee fails to understand what the authors meant here.

When RSNO “release” •NO (reaction 2) or NO+ (Reaction 1) there is no disulfide bond formed as a direct product, while when the disulfide bond is formed, the specie released is HNO (reaction 3). Please, see the reactions below:

RSNO + R’SH => RSH + R’SNO (transnitrosation)  (reaction 1)

RSNO => RS• + •NO (homolytic bond break)        (reaction 2)

RSNO + R’SH => RSSR’ + HNO                                   (reaction 3)

Lines 192-195 – The authors generalize the reaction mechanism of GSNO denitrosation by carbonyl reductase 1, alcohol dehydrogenase (ADH5), and thioredoxin/thioredoxin reductase. However, the thioredoxin/thioredoxin reductase system does not produce ammonia. Please, revise this.

Minor:

Line 52 –  Change “…beneficial aspects of •NO produced by NO” to “…beneficial aspects of •NO produced by NOS”

Author Response

The manuscript reviews the biological production and reactions of nitric oxide (NO). It is also discussed the influences of NO production in the tricarboxylic acid and urea cycles. This theme is interesting and may attract the interest of scientists working in the area of NO. Despite that, the manuscript could benefit from some corrections.

Comments:

Line 38 – “Peroxynitrite (ONOO–), which is formed from •NO and O2•−, is the most powerful oxidant and may be responsible for oxidative damage to surrounding tissues”. This sentence needs revision. The •OH is the most powerful (and less selective) known biological oxidant, not peroxynitrite.

Responses:

Thank you for pointing out. We have rephrased the sentence by replacing “the most” to “a” as follows.

“Peroxynitrite (ONOO), which is formed from •NO and O2, is a powerful oxidant and may be responsible for oxidative damage to surrounding tissues [7].”

In the section “Iron as a target of •NO” the authors correctly affirm that heme and non-heme iron are preferential targets of •NO. However, the non-heme includes not only the iron-sulfur clusters but also the labile iron pool (LIP). LIP is an important NO target and should be discussed in this section.

Responses:

Thank you for kind advice. We understand importance of the labile iron pool (LIP) as the target of NO. We have added following paragraph about LIP. 

“While carbon monoxide and oxygen molecule bind ferrous iron that is mostly present in the form of heme, •NO exceptionally binds ferric iron too [18]. The labile iron pool (LIP), which is defined as the iron fraction chelatable by high-affinity metal chelators, accounts for 0.1-3% of the total cellular iron. Reaction of •NO with LIP produces stable dinitrosyl iron complexes, which are the most abundant adduct and may act as •NO carriers within cells [20, 21]. ”

[new references]

  1. Toledo, J.C. Jr.; Bosworth, C.A.; Hennon, S.W.; Mahtani, H.A.;, Bergonia, H.A.; Lancaster, J.R. Jr. Nitric oxide-induced conversion of cellular chelatable iron into macromolecule-bound paramagnetic dinitrosyliron complexes. J. Biol. Chem. 2008, 283, 28926-28933. doi: 10.1074/jbc.M707862200.
  2. Hickok, J.R.; Sahni, S.; Shen, H.; Arvind, A.; Antoniou, C.;, Fung, L.W.; Thomas, D.D. Dinitrosyliron complexes are the most abundant nitric oxide-derived cellular adduct: biological parameters of assembly and disappearance. Free Radic. Biol. Med. 2011, 51, 1558-1566. doi: 10.1016/j.freeradbiomed.2011.06.030.

Lines 101-103 “The oxidative inactivation of aconitase releases one ferrous iron unit along with hydrogen peroxide, which results in the formation of [3Fe-4S] [21].” In this sentence the authors do not identify the oxidant that leads to aconitase inactivation, therefore it is not possible to affirm that hydrogen peroxide is a byproduct. Please, revise this information.

 Responses:

We have revised the sentence by adding oxidant species as follows. “The oxidative inactivation of aconitase by anionic oxidants such as O2 and ONOO releases one ferrous iron unit along with hydrogen peroxide, which results in the formation of [3Fe-4S]”.

In the section “Peroxynitrite as a potent oxidant” the authors neglect the peroxynitrite’s pKa (6.8) and also the difference in reactivity of ONOOH and ONOO- species. The ONOOH decay leads to nitrate, •NO2 and •OH, not ONOO- decay. Please, correct this information (lines 144-148).

Responses: Thank you very much for kindly pointing out. We have corrected the statement about decomposition of ONOO- and ONOOH and also corrected the issue in Fig. 2.

Line 177 – The authors should better explain why the RSNO formation by •NO is a slow process. Which mechanism are they referring to?

Responses:

We could not find other reports on mammalian SNO synthase in the literature after ref [60], so it is a misinterpretation of the content.In fact, as indicated in many publications, this reaction is rather efficient process. So the presence of the enzyme may further stimulate it.Therefore, the corresponding description has been revised as follows.

“While the RSNO formation by •NO is efficient process, enzymes designated as SNO synthases may accelerate the formation of SNO as reported in E. coli [58].”

Line 179 – “Because RSNO may release bound •NO or NO+ and leave the disulfide bond in the original compound, it is not an end product but is regarded as a •NO donor.”

This referee fails to understand what the authors meant here.

When RSNO “release” •NO (reaction 2) or NO+ (Reaction 1) there is no disulfide bond formed as a direct product, while when the disulfide bond is formed, the specie released is HNO (reaction 3). Please, see the reactions below:

RSNO + R’SH => RSH + R’SNO (transnitrosation)  (reaction 1)

RSNO => RS• + •NO (homolytic bond break)        (reaction 2)

RSNO + R’SH => RSSR’ + HNO                                   (reaction 3)

Responses: Thank you for kindly pointing out the incorrect description and providing us valuable information concerning the reactions. According to them, we have corrected the statement as follows.

“RSNO undergoes several conversion reactions depending on the reaction conditions [61]. Homolytic bond break converts RSNO to RS• + •NO. Reaction of R1SNO and the other thiol compound (R2SH) produces either R1SH + R2SNO by transnitrosation or R1SSR2 + HNO.”

  1. Fukuto, J.M.; Perez-Ternero, C.; Zarenkiewicz, J.; Lin, J.; Hobbs, A.J.; Toscano, J.P. Hydropersulfides (RSSH) and Nitric Oxide (NO) Signaling: Possible Effects on S-Nitrosothiols (RS-NO). Antioxidants (Basel). 2022, 11, 169. doi: 10.3390/antiox11010169.

Lines 192-195 – The authors generalize the reaction mechanism of GSNO denitrosation by carbonyl reductase 1, alcohol dehydrogenase (ADH5), and thioredoxin/thioredoxin reductase. However, the thioredoxin/thioredoxin reductase system does not produce ammonia. Please, revise this.

Responses: Thank you for kindly pointing out the mistake in the description. We have corrected corresponding statements as follows.

 “These enzymes include carbonyl reductase 1 (CBR1) and class III alcohol dehydrogenase (ADH5) [50,71]. On the other hand, thioredoxin/thioredoxin reductase systems degrades GSNO to •NO + GSH or GSSG.”

Minor:

Line 52 –  Change “…beneficial aspects of •NO produced by NO” to “…beneficial aspects of •NO produced by NOS”

Responses: Responses: Thank you for pointing out the typo. We have corrected the second NO to NOS.

Reviewer 3 Report

Dear authors,

Topic of your manuscript is interesting and it may be of interest for the readers of Molecules. However, I have several suggestions/comments on how you should revise the manuscript prior to the potential publication.

1. Title should be revised as it does not reflect precisely manuscript's messages. See last sentence of the abstract: Herein we overview the protective aspects of •NO against radical species and the autoregulatory systems that are enabled by the metabolic remodeling in M1-polarized macrophages. This sentence may guide you in defining better and content-related title. 

2. You should avoid using same words in keywords and title in order to increase visibility of your manuscript. 

3. Please consider someone who is not an expert in your field reading your manuscript. It would be very hard for him/her to grasp the introduction. Especially first part of the introduction. Please introduce readers to EDRF. In Figure 1 please correct "down regulation" to "down-regulation".

4. It is not clear what do you add to already existing literature by your work. Please emphasize this throughout the text. Try to comment on existing knowledge instead of listing well-known facts.

5. Chapter Perspectives is rather a summary of the paper. Please consider revising.

Author Response

Topic of your manuscript is interesting and it may be of interest for the readers of Molecules. However, I have several suggestions/comments on how you should revise the manuscript prior to the potential publication.

  1. Title should be revised as it does not reflect precisely manuscript's messages. See last sentence of the abstract: Herein we overview the protective aspects of •NO against radical species and the autoregulatory systems that are enabled by the metabolic remodeling in M1-polarized macrophages. This sentence may guide you in defining better and content-related titl

Responses:

Thank you for kind advice. We have changed title as follows.

Involvement of nitric oxide in protecting against radical species and autoregulation of M1-polarized macrophages through metabolic remodeling

  1. You should avoid using same words in keywords and title in order to increase visibility of your manuscript. 

Responses:

Thank you for kind advice. We have changed keywords.

NOS2; tricarboxylic acid cycle; urea cycle; aconitase; polyamines.

  1. Please consider someone who is not an expert in your field reading your manuscript. It would be very hard for him/her to grasp the introduction. Especially first part of the introduction. Please introduce readers to EDRF. In Figure 1 please correct "down regulation" to "down-regulation".

Responses: Thank you for kind advice. We have added explanatory sentence to Introduction as follows.

“Endothelial cells play an essential role in acetylcholine-induced relaxation of the vasculature.”

We have corrected "down regulation" to "down-regulation" in Figure 1.

  1. It is not clear what do you add to already existing literature by your work. Please emphasize this throughout the text. Try to comment on existing knowledge instead of listing well-known facts.

Responses: Thank you for kind advice. This is a review article, so all observations have been reported already. We have cited three our original works (ref No. 74, 88 and 113). We are citing them because they provide required information on corresponding parts, just like other papers, but not citing arbitrarily. According to your advices, now, we have added brief comments on them (ref 74 and 88) where cited.

  1. Chapter Perspectives is rather a summary of the paper. Please consider revising.

Responses: Thank you for kind advice. We have revised Perspectives chapter to clarify current issues.

Round 2

Reviewer 3 Report

Dear authors,

Thank you for revising the manuscript.